# Multifractal Analysis to Determine the Effect of Surface Topography on the Distribution, Density, Dispersion and Clustering of Differently Organised Coccal-Shaped Bacteria

**DOI:** 10.3390/antibiotics11050551

**Published:** 2022-04-21

**Authors:** Adele Evans, Anthony J. Slate, Millie Tobin, Stephen Lynch, Joels Wilson Nieuwenhuis, Joanna Verran, Peter Kelly, Kathryn A. Whitehead

**Affiliations:** 1Faculty of Science and Engineering, Manchester Metropolitan University, Chester Street, Manchester M1 5GD, UK; adele.evans@mmu.ac.uk (A.E.); j.verran@mmu.ac.uk (J.V.); peter.kelly@mmu.ac.uk (P.K.); 2Department of Biology and Biochemistry, University of Bath, Claverton Down, Bath BA2 7AY, UK; ajs319@bath.ac.uk; 3Department of Computing and Mathematics, Manchester Metropolitan University, Chester Street, Manchester M1 5GD, UK; millietobin@gmail.com (M.T.); s.lynch@mmu.ac.uk (S.L.); 4Microbiology at Interfaces, Manchester Metropolitan University, Chester Street, Manchester M1 5GD, UK; joelswn@hotmail.co.uk

**Keywords:** coccal bacteria, multifractal analysis, linear topography, density, dispersion, clustering

## Abstract

The topographic features of surfaces are known to affect bacterial retention on a surface, but the precise mechanisms of this phenomenon are little understood. Four coccal-shaped bacteria, *Staphylococcus sciuri*, *Streptococcus pyogenes*, *Micrococcus luteus*, and *Staphylococcus aureus*, that organise in different cellular groupings (grape-like clusters, tetrad-arranging clusters, short chains, and diploid arrangement, respectively) were used. These differently grouped cells were used to determine how surface topography affected their distribution, density, dispersion, and clustering when retained on titanium surfaces with defined topographies. Titanium-coated surfaces that were smooth and had grooved features of 1.02 µm-wide, 0.21 µm-deep grooves, and 0.59 µm-wide, 0.17 µm-deep grooves were used. The average contact angle of the surfaces was 91°. All bacterial species were overall of a hydrophobic nature, although *M. luteus* was the least hydrophobic. It was demonstrated that the 1.02 µm-wide featured surface most affected *Strep. pyogenes* and *S. sciuri*, and hence the surfaces with the larger surface features most affected the cells with smaller dimensions. The 0.59 µm featured surface only affected the density of the bacteria, and it may be suggested that the surfaces with the smaller features reduced bacterial retention. These results demonstrate that the size of the topographical surface features affect the distribution, density, dispersion, and clustering of bacteria across surfaces, and this is related to the cellular organisation of the bacterial species. The results from this work inform how surface topographical and bacterial properties affect the distribution, density, dispersion, and clustering of bacterial retention.

## 1. Introduction

Work has been carried out to determine the effect of surface topography [1], chemistry [2], and physiochemistry [3] on bacterial retention, but the results are often conflicting. The main parameter often associated with bacterial retention is surface roughness [4], which is a measure of the texture of a surface. Peaks and troughs are measured using a profilometer, whereby *R_a_* (roughness average) is the arithmetic average of the absolute values of the profile height deviations from the mean. It is suggested that a rougher surface topography may result in enhanced bacterial retention, which is deemed more beneficial for biofilm formation [5,6]. The average size of bacterial cell length ranges from 0.2 to 2.0 µm for cocci and 1.5 to 5.0 μm for rod-shaped bacteria, and it has been suggested that topographical surface features in a region similar to that of the bacterial cell dimensions results in enhanced bacterial attachment [7,8]. In 2005, Whitehead et al. demonstrated that surfaces with a defined size (0.2, 0.5, 1.0, and 2.0 μm) had an effect on bacterial retention, whereby *Staphylococcus aureus* (cocci-shaped; 0.5–1.0 µm diameter) were retained in 0.5 μm-sized features, whilst *Pseudomonas aeruginosa* (rod shaped; 1.0–3.0 μm diameter) were preferentially retained, often end-on, within 1.0 μm features [9]. This study demonstrated that surface features can enhance or impede cell retention depending on whether the feature is larger or smaller than the cell diameter. Furthermore, a recent study by Caro-Lara et al. (2021) utilised microtopographic texturization using direct laser interference patterning to texture copper with linear patterns at 4.7 µm, 6.8 µm, 14 µm, and 18 µm periodically [10]. Bacterial adhesion was characterised using a copper-resistant bacterium, *Variovorax* sp., and it was demonstrated that a periodic interval of 4.7 µm was the most efficient pattern to suppress *Variovorax* sp. initial adhesion when compared to the nontextured surface [10].

Bacterial attachment to abiotic surfaces is a cause for concern in many industries, including both the medical and food industries. Bacterial attachment to abiotic materials often results in biofilm formation, which can lead to the contamination of biomedical devices and food products [11]. The presence of viable bacterial cells (and their products) on food contact surfaces can result in product spoilage and infection. Bacterial surface interactions and therefore bacterial attachment, adhesion, and subsequent biofilm formation are poorly understood. This is due to the combined and complex interplay of a myriad of factors including (but not limited to) surface roughness, physicochemistry, the composition of a surface (including charge and heterogeneity), bacterial cell surface hydrophobicity and cell surface charge, outer membrane proteins, and the presence of bacterial cell appendages (e.g., type IV pili) [11,12,13,14,15,16].

Linear surface features regularly occur on food-contact surfaces through general use. As well as enhancing bacterial retention, the presence of such surface features can reduce the cleanability of a surface, providing protection against shear forces and cleaning regimes. Contact surfaces are an important source of bacterial transmission [17]. Whilst some bacteria are regarded as transient organisms, others may establish themselves, forming residential communities on surfaces in industrial environments [18]. Biofilm formation occurs after initial bacterial attachment and adhesion and confers many advantages to biofilm-associated cells, such as resistance to physical, mechanical, and chemical forces [19,20].

In this study, the retention of four different coccal-shaped bacterial species, *Staphylococcus sciuri*, *Streptococcus pyogenes*, *Micrococcus luteus*, and *Staphylococcus aureus*, that organise in different cellular groupings were compared when retained on surfaces with defined chemistry, topography, and physicochemistry. The results from this work will inform how surface and bacterial properties affect the distribution, density, dispersion, and clustering of bacterial retention. The novelty of this work is that most work evaluates the effect of surface topography based on cellular size, rather than on bacterial groupings.

## 2. Methods

### 2.1. Surface Fabrication

The surfaces utilised throughout this study had previously been developed and characterised [21]. Briefly, mirror-finished polished silicon wafers (Montco Silicon Technologies, San Jose, CA, USA) were used to produce smooth titanium surfaces and were coated with titanium via magnetron sputtering. Digital versatile discs (DVDs) and compact discs (CDs) were soaked overnight in 30% sodium hydroxide (BDH, Poole, UK), then for 30 min in sterile distilled water, rinsed, and then dried, thus removing the protective coating and exposing the bare substratum. All surfaces were prepared as 1 cm^2^ sections prior to use. Magnetron sputtering was conducted to deposit titanium onto the substrates in a modified Edwards E306A coating system rig using a single 150 mm diameter × 10 mm thick, 99.5% pure titanium target with the substrates biased at −50 V. The surfaces were removed from the rig and placed into sterile Petri dishes, in which they were stored at room temperature until use.

### 2.2. Surface Characterisation

The surfaces were characterised in terms of their topography and wettability according to Verran et al. [22]. Atomic force microscopy (Veeco, St Ives, UK) was operated in contact mode to characterise the topography of the surfaces. Silicon nitride tips with a spring constant of 0.12 N/m were used. The RMS values (defined as the square root of the mean square) were derived from replicate readings (*n* = 3). Contact angle measurements were taken using the sessile drop technique with 5 µL of water (Kruss Goniometer, Toulouse, France) using HPLC-grade water (BDH, Poole, UK) (*n* = 5).

### 2.3. Preparation of Microorganisms

*Staphylococcus sciuri* strain CCL 101, *Micrococcus luteus* strains ATCC 15307, NCIMB 9278, NCTC 2665, *Streptococcus pyogenes* strain NCTC 12696, and *Staphylococcus aureus* strain NCIMB9518 were streaked onto nutrient agar (NA) (Oxoid, Basingstoke, UK) and grown overnight at 37 °C. One hundred millilitres of nutrient broth (NB) (Oxoid, Basingstoke, UK) was inoculated with a single colony of bacteria and incubated at 37 °C for 18 h. The bacteria were harvested at 716× *g* for 10 min, washed thrice in sterile distilled water, and then centrifuged at 716× *g* for 10 min. Cells were resuspended to an optical density (OD) of 1.0 ± 0.1 at 540 nm in sterile distilled water. Cell enumeration was calculated via colony-forming units (CFU mL^−1^), and the initial cell suspensions were ca. 5.0 × 10^8^ CFU mL^−1^.

### 2.4. Retention Assays

The titanium-coated surfaces were placed in sterile Petri dishes to which 25 mL of standardised cell suspension was added. The surfaces were incubated horizontally at 37 °C without agitation for 1 h. After the incubation period, the surfaces were removed with sterile tweezers, and each surface was washed gently along the grooves, once with 5 cm^3^ distilled H_2_O, with the distilled water bottle at a 45° angle, and with a 3 mm nozzle. Substrata were placed in a laminar flow hood and air-dried. The samples were prepared for scanning electron microscopy (SEM). Ten images were taken per surface, and biological repeats were carried out in triplicate (*n* = 3).

### 2.5. Scanning Electron Microscopy

The samples with bacterial cells were immersed in 4% *v*/*v* gluteraldehyde for 24 h at 4 °C. After fixing, the samples were thoroughly rinsed with 100 cm^3^ distilled H_2_O. The samples were subjected to an ethanol gradient (30%, 50%, 70%, 90%, and 100% *v*/*v* absolute ethanol), at 10 min per ethanol concentration. Prior to examination, the samples were stored at room temperature in a phosphorous pentoxide (Sigma–Aldrich, Gillingham, UK) desiccator. The samples were fixed to stubs for gold sputter coating, which was carried out using a Polaron E5100 (Polaron CVT Ltd., Milton Keynes, UK) SEM sputter coater. Samples were sputter-coated at a vacuum of 0.09 mbar, for 3 min, at 2500 V, in argon gas at a power of 18–20 mA. Images of substrata were obtained using a JEOL JSM 5600LV scanning electron microscope (Jeol Ltd., Welwyn Garden City, UK).

### 2.6. Multifractal Analysis

Multifractal analysis of the SEM images (cells and surfaces) was conducted in a manner similar to that adopted by Wickens et al. [22]. Briefly, Figure 1 shows the motifs used to produce multifractals (left column), representative figures (central column), and the computed f-α curves (right column), in three cases. The top figures show a multifractal with clusters of dark pixels, and the red f-α curve is skewed left (Δf < 0). The central figures display the motif and multifractal image for a homogeneous multifractal (low dispersion and clustering), leading to a narrow red f-α curve, which is neither skewed left nor right (Δf≈0). The bottom row depicts a multifractal with clusters of bright pixels, and the red f-α curve is skewed right (Δf>0).

In this paper, the numerical *f*(α) spectra were computed for −10 ≤ q ≤ 10 in all cases, and data set boxes of size ε = 4, 8, 16, 32, 64, 128, and 256 were used in the computation. From this, ƒ(α) curves were generated, and this enabled relative density (height of the f-α curve), dispersion (width of the f-α curve), and clustering (Δf) to be computed and compared between the different surfaces with bacterial species.

### 2.7. Microbial Adhesion to Hydrocarbons (MATH)

In order to determine the physicochemistry of the bacterial species, a MATH assay was carried out. The bacteria were grown in NB at 37 °C overnight and centrifuged for 10 min at 2210× *g*. The bacteria were washed three times in PUM buffer (pH 7.1, K_2_HPO_4_·3H_2_O 22.2 g/L, KH_2_PO_4_ 7.26 g/L, urea 1.8 g/L, and MgSO_4_·7H_2_O 0.2 g/L) and were re-suspended at 400 nm to an OD of 1.0. The washed bacterial suspension (1.5 mL) was added to a 15 mm diameter, round bottomed test tube, to which 250 µL of a solvent (ethyl acetate, chloroform, decane, hexadecane (BDH, Poole, UK)) was added. The suspensions were incubated at 37 °C for 10 min, vortexed for 2 min, and incubated at 37 °C for 30 min. Following removal, the OD was determined at 400 nm of the lower aqueous phase. The results were calculated to determine cell surface hydrophobicity Rosenberg et al. [23].
$$\mathrm{adhesion}=(\frac{1-A}{Aø})\times 100$$

*A*ø is the OD of the bacterial suspension before the addition of the solvent; *A* is the OD of the lower aqueous phase after mixing with the solvent (*n* = 3).

### 2.8. Statistical Analysis

Gaussian distribution was assumed following Shapiro–Wilk normality analysis. Statistical analysis was conducted by performing two-way ANOVA, coupled with Tukey’s multiple comparison tests for post hoc analysis using GraphPad Prism (version 8.4.3; GraphPad Software, San Diego, CA, USA) in order to determine significant differences at a confidence level of 95% (*p* < 0.05). Error bars represent the standard error of the mean. Asterisks denote significance, * (*p* ≤ 0.05), ** (*p* ≤ 0.01), *** (*p* ≤ 0.001), and **** (*p* ≤ 0.0001).

## 3. Results

The surfaces in this work were titanium-coated using physical vapour deposition to create surfaces with a regular chemistry (Figure 2A–C) and which included those with defined grooved features of 1.02 µm-wide, 0.21 µm-deep grooves (Figure 3a), and 0.59 µm-wide, 0.17 µm-deep grooves (Figure 3b).

### 3.1. Surface Characterisation

The shape of the surface features was quantified using AFM to determine the RMS of the samples (Figure 4). It was determined that there was a significant difference in the RMS value between each of the three surfaces used (*p* > 0.001). According to contact angle measurements, no significant variation (*p* < 0.001) in surface wettability was observed, with an average contact angle of 91° being determined for all three samples (data not shown).

### 3.2. Distribution of Bacteria across the Surfaces

SEM images of *Staphylococcus sciuri*, *Streptococcus pyogenes*, *Micrococcus luteus*, and *Staphylococcus aureus* were produced to quantify the distribution, density, dispersion, and clustering of the microbial cells on the surfaces of the 1.02 µm-wide featured surfaces (Figure 5A–D), 0.59 µm-wide featured surfaces (Figure 6A–D), and the smooth surfaces (Figure 7A–D). The images revealed the different groupings of bacteria on the surface. The diameter of the bacteria was approximately 0.8 µm for *S. sciuri*, 0.9 µm for *Strep. pyogenes*, 1.2 µm for *M. luteus,* and 0.9 µm for *S. aureus*.

It was apparent from the images that the different bacteria formed their natural groupings on the surfaces of the different topographies. *S. aureus* clustered in groups within which grape-like clusters could be seen (Figure 5, Figure 6 and Figure 7A), whereas the *M. luteus* clustered in clear tetrad-arranging clusters (Figure 5, Figure 6 and Figure 7B). The *Strep. pyogenes* arranged in short chains (Figure 5, Figure 6 and Figure 7C), and *S. sciuri* generally arranged in small clusters or a diploid arrangement (Figure 5, Figure 6 and Figure 7D).

### 3.3. Density of Bacteria across the Surfaces

On the 1.02 µm-wide featured surface, there was only one significant difference in the density of the bacteria across the surfaces, between the *Strep. pyogenes* (1.16) and *S. sciuri* (1.46) (Figure 8A). On the 0.59 µm featured surface, there were significant differences between the bacterial density of all the surfaces, except for *S. aureus* (1.46) and *M. luteus* (1.49), which were in larger clumps and hence less affected. On the smooth surface, which would represent the typical cellular arrangement, there was a relationship demonstrated with respect to the density of the bacteria and the bacterial species with *S. aureus* (1.48), *M. luteus* (1.25), *Strep. pyogenes* (1.04), and *S. sciuri* (0.96) decreasing in density. Significant differences were observed in the results between the *S. aureus* and the *Strep. pyogenes* and *S. sciuri* on the smooth surfaces.

### 3.4. Dispersion of Bacteria across the Surfaces

It was determined that on the 1.02 µm-wide featured surface, there was only one significant difference in the results between the *S. aureus* (1.34) and *S. sciuri* (0.98), highlighting the compactness of the bacteria across the surfaces (Figure 8B).

### 3.5. Clustering of Bacteria across the Surfaces

Among the 1.02 µm-wide featured surfaces, there were significant differences in the bacteria on the surfaces, except for *S. aureus* (0.69) and *M. luteus* (0.61), which were similar (Figure 8C). On the 0.59 µm featured surface, there was no significant difference in clustering of the bacteria across the surfaces. On the smooth surface, there was a significant difference in all the bacterial clusterings.

### 3.6. Microbial Adhesion to Hydrocarbons

All the bacterial species demonstrated a high affinity for the apolar n-alkanes decane and hexadecane, demonstrating that they were overall of a hydrophobic nature, although *M. luteus* was the least hydrophobic of these (91.2 chloroform and 82.9 hexadecane) (Figure 9). All the bacteria were also basic (Lewis-basic), as demonstrated by their affinity for the solvent chloroform. The only significant difference in the affinity of the bacterial species to the solvents was demonstrated for the basic (Lewis-basic) solvent ethyl acetate, whereby *S. sciuri* demonstrated the lowest affinity (9.7), followed by *M. luteus* (11.9), *Strep. pyogenes* (68.6), and *S. aureus* (91.7). The greater the bacterial affinity for the pair chloroform and hexadecane indicated that these bacteria were better electron donators, with *M. luteus* and *S. sciuri* being stronger electron donators due to their low affinity to ethyl acetate. For those species that had a higher combined affinity for the n-polar (decane and hexdecane) vs. polar (chloroform and ethyl), the hydrophobic properties are stronger than the electron properties.

## 4. Discussion

Although the effect of surface topography has been explored to understand how it affects bacterial retention to surfaces [9], this phenomenon is still little understood. This work used surfaces with defined chemistry and wettabilities to determine the effect surface topography had on bacterial retention. Gram-positive, coccal-shaped bacteria of similar sizes were selected for use in these assays. However, these bacterial species naturally form different groupings (e.g., pairs, chains. tetrads, grape-like clusters) when visualised microscopically.

### 4.1. Microbial Adhesion to Hydrocarbons (MATH)

The MATH assay demonstrated that all the bacterial species were hydrophobic and that the *M. luteus* and *S. sciuri* were stronger electron donators. However, the physicochemical nature of the bacterial species in this instance did not seem to influence the distribution, density, dispersion, or clustering of the bacteria across the surfaces.

### 4.2. Distribution of Bacteria across the Surfaces

All the bacteria were observed in their expected groupings on the surfaces. On the smooth titanium surfaces, *S. sciuri* was in pairs or clusters, *Strep. pyogenes* was observed as pairs or in chains touching end to end [24], *M. luteus* was in a typical tetrad formation, and *S. aureus* was predominantly in grape-like clusters. The images confirmed the presence of bacterial sizes on the smooth titanium with dimensions in agreement with the literature [25,26,27,28]. Following multifractal analysis of the bacteria retained on the surfaces, the results demonstrated that the size of topographical features affected the distribution, density, dispersion, and clustering of the bacteria across surfaces, and this was related to the organisation of the bacterial species.

It was demonstrated that the 1.02 µm-wide featured surface most affected *Strep. pyogenes* and *S. sciuri*, and hence the surfaces with the larger surface features most affected the cells with the smaller dimensions. The 0.59 µm featured surface only affected the density of the bacteria, and it may be suggested that the surfaces with the smaller features reduced bacterial retention. The smooth surfaces, which were used to demonstrate the natural cellular organisation of the cells, affected the clustering of all the bacteria, which was to be expected. The smooth surface also affected the density of *S. aureus*, *Strep. Pyogenes*, and *S. sciuri.* Work by others has used multifractal analysis to determine the spatial organisation of bacteria in a range of scenarios. Chavez et al. (2011) demonstrated that the use of multifractal analysis helped to design a sensor that would provide a more efficient control of plant disease through an early and full spatial assessment of the health status of the crop [29]. Statistical analysis has also been used to determine an individual-based model to explain observed variation in the size of bacterial clusters on plant leaf surfaces to understand how different ‘waterscapes’ impacted the diffusion of nutrients from a leaf interior to the surface and the growth of individual bacteria on these nutrients [30]. Surface conditioning with cellular components has been demonstrated to affect the amount and clustering of bacteria on polystyrene surfaces and their propensity to induce biofilm formation [31]. With respect to the effect of the surfaces on bacterial clustering, it was shown that soft surfaces resulted in more microcolonies than hard ones, and this was explained due to the proportion of motile cells [32]. The results presented in this work demonstrate that the size of topographical surface features affect the distribution, density, dispersion, and clustering of bacteria across surfaces, and this is related to the organisation of the bacterial species. This information clarifies that bacterial species are not retained on surfaces in the same patterns, even when the bacterial species are the same shape.

When more diverse bacterial species are used, both in shape and cell wall constituents (Gram staining), the findings of the effect of the surface properties on bacterial retention are also different. For example, previous work has found that the retention of microbial cells can both be enhanced or impeded based on the relationship between the surface topography and size and shape of the cells [33]. Titanium-coated surfaces with a grooved topography (0.59 and 1.02 µm) reduced the adherence of the rod-shaped Gram-negative bacteria *Escherichia coli* but were found to show an increased retention on the surface features similar to its cell dimensions, for example, using Gram-positive, coccal-shaped *Staphylococcus sciuri* [21]. If bacteria of similar sizes are used with topographical and chemically different surfaces, different surface properties are shown to affect the results. For example, when multifractal analysis was applied to the bacteria retained on stainless steel and titanium nitride-silver surfaces, which had used bacteria with differently shapes and cell wall constituents (Gram-negative, rod-shaped *Escherichia coli* and Gram-positive, coccal-shaped *Staphylococcus sciuri*), it was demonstrated that the physicochemistry of the bacteria and surfaces both influenced bacterial retention, and multifractal analysis of the retained bacteria demonstrated that the surface properties affected the spread and clustering, but not the density of the bacteria [34].

In most assays, different topographies are made on a range of surfaces that result in a variety of surface-property combinations, which also results in the surfaces having a range of different chemistries and hydrophobicities [35]. Surfaces with a range of properties have been shown to elicit a range of effects both increasing and decreasing bacterial adhesion. On titanium implant surfaces, it was demonstrated that an *R_a_* value of 0.2 μm and below did not influence the quantity of bacterial adhesion [36]. For example, it has been suggested that rougher surfaces support bacterial colonization [37,38], although the results are conflicting. Early biofilm formation by *Streptococcus mutans* has also been shown to be reduced on conventionally polished surfaces when compared to surfaces polished with a simplified protocol which removed less material from the surface, and hence it was concluded that polishing did not necessarily reduce oral bacterial colonisation [39]. Differences in the bacterial species can also affect their binding to a surface. It was shown that titanium surfaces showed higher *Streptococcus sanguinis* adhesion on rough surfaces when they were compared to surfaces with a medium and smooth finish, and although there was a trend, the results were not statistically significant [40]. However, the use of titanium surfaces was found not to influence the quantity of adhering *S. epidermidis* [40]. Such results demonstrate how the bacterial species can also affect their retention to surfaces. However, using such surfaces, it is difficult to determine how specific surface parameters affect bacterial retention directly. Using very controlled surfaces, produced using the same materials and deposition times, as was used in this work, results in such interactions can be investigated.

Although there has been much debate regarding how surface topography affects bacterial retention on surfaces, when using surfaces whereby only the topographical features vary, the results from this study suggest that differences in cell shape and cell wall constituents (Gram staining) affect the distribution, density, dispersion, and clustering of bacteria on surfaces of defined topographies.

## 5. Limitations of the Study

The limitations of this study include that we only investigated linear surface features that are of microbial sizes in attempt to understand the topographical effect on microorganisms with different groupings. However, further work to investigate the different chemistries and hydrophobicities will also be of value in future work.

## 6. Conclusions

All bacterial species were overall of a hydrophobic nature, although *M. luteus* was the least hydrophobic. It was demonstrated that the 1.02 µm-wide featured surface most affected *Strep. pyogenes* and *S. sciuri*, and hence the surfaces with the larger surface features most affected the cells with the smaller dimensions. The 0.59 µm featured surface only affected the density of the bacteria, and it may be suggested that the surfaces with the smaller features reduced bacterial retention. The smooth surfaces, which were used to demonstrate the natural cellular organisation of the cells, affected the clustering of all the bacteria. These results demonstrate that the size of topographical surface features affect the distribution, density, dispersion, and clustering of bacteria across surfaces, and this is related to the organisation of the bacterial species. This information clarifies that bacterial species are not retained on surfaces in the same patterns, even when the bacterial species are the same shape.

## Figures and Tables

**Figure 1 antibiotics-11-00551-f001:**
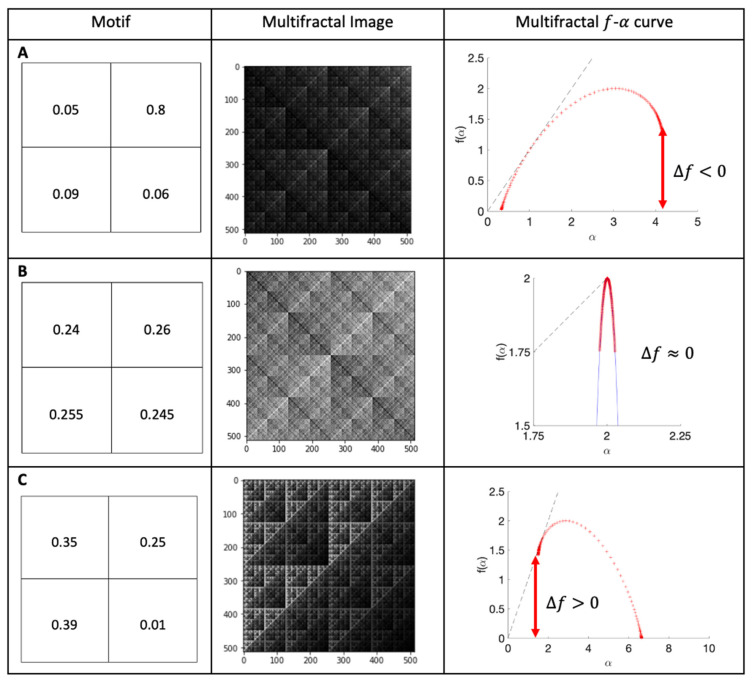
Computed curves for theoretical multifractal images. Motif, multifractal image and computed curves: (**A**) clusters of dark pixels; (**B**) homogeneous multifractal; (**C**) clusters of bright pixels.

**Figure 2 antibiotics-11-00551-f002:**
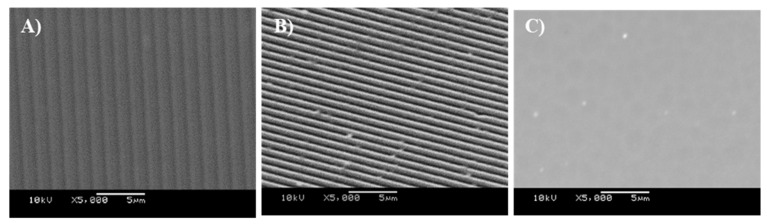
Scanning electron microscopy (SEM) images of (**A**) Ti-coated CD surface, (**B**) Ti-coated DVD surface, and (**C**) TiSi surface.

**Figure 3 antibiotics-11-00551-f003:**
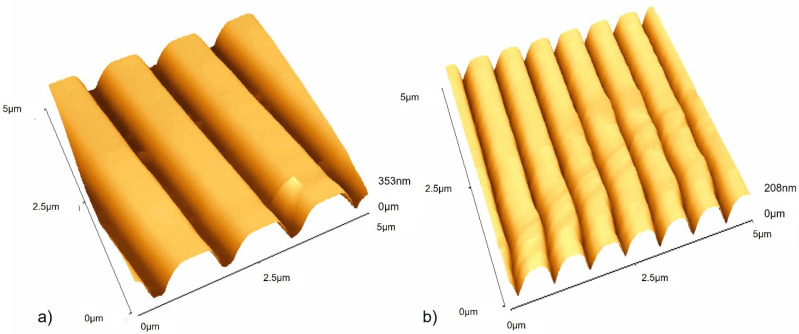
Atomic force microscopy (AFM) images of (**a**) Ti-coated CD surface and (**b**) Ti-coated DVD surface.

**Figure 4 antibiotics-11-00551-f004:**
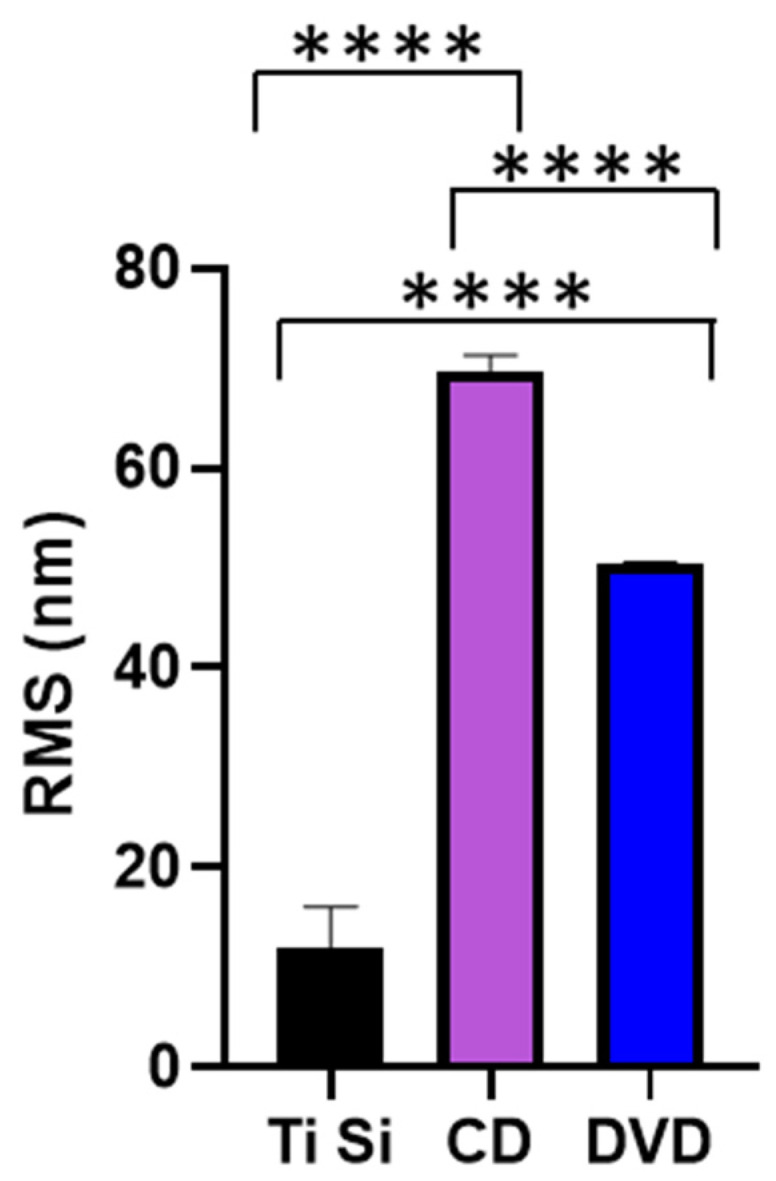
RMS values of the titanium-coated surfaces. **** *p* < 0.001.

**Figure 5 antibiotics-11-00551-f005:**
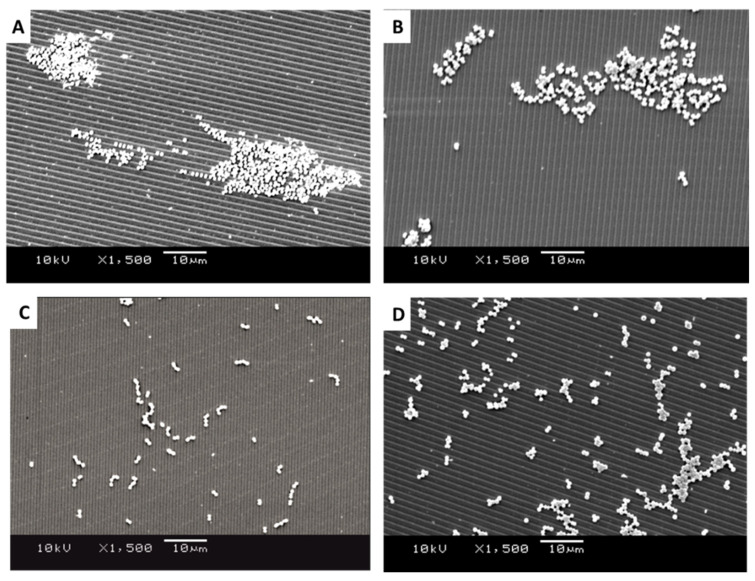
SEM images of retained bacteria on silica plates with titanium-coated grooves (1.02 µm): (**A**) *S. aureus*, (**B**) *M. luteus*, (**C**) *Strep. pyogenes*, and (**D**) *S. sciuri*.

**Figure 6 antibiotics-11-00551-f006:**
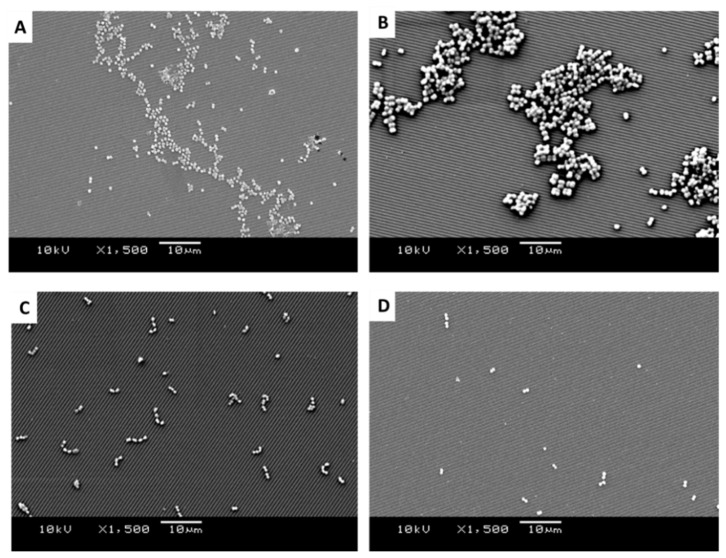
SEM images of retained bacteria on silica plates with titanium-coated grooves (0.59 µm): (**A**) *S. aureus*, (**B**) *M. luteus*, (**C**) *Strep. pyogenes*, and (**D**) *S. sciuri*.

**Figure 7 antibiotics-11-00551-f007:**
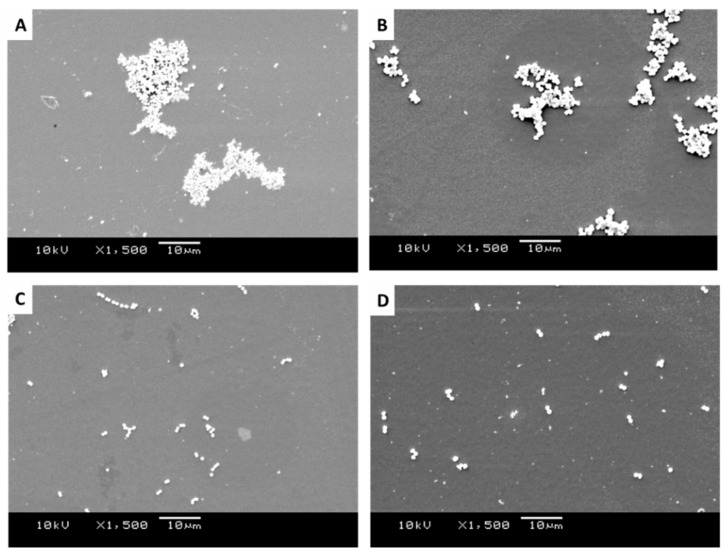
SEM images of retained bacteria on smooth titanium: (**A**) *S. aureus*, (**B**) *M. luteus*, (**C**) *Strep. pyogenes*, and (**D**) *S. sciuri*.

**Figure 8 antibiotics-11-00551-f008:**
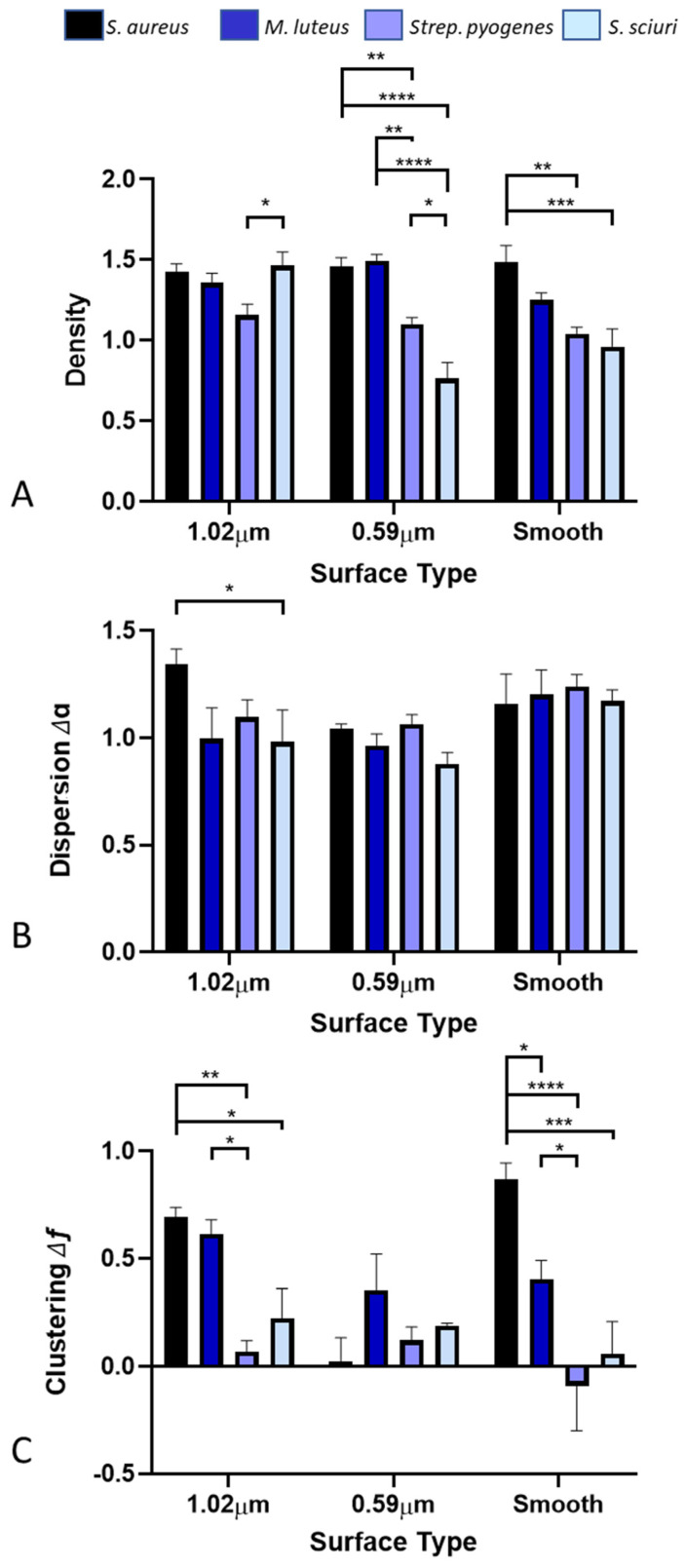
Density (**A**), dispersion (**B**), and clustering (**C**) of the bacterial species on the different surfaces determined via multifractal analysis. Stars denote * *p* < 0.05, ** *p* < 0.01, *** *p* < 0.005, **** *p* < 0.001.

**Figure 9 antibiotics-11-00551-f009:**
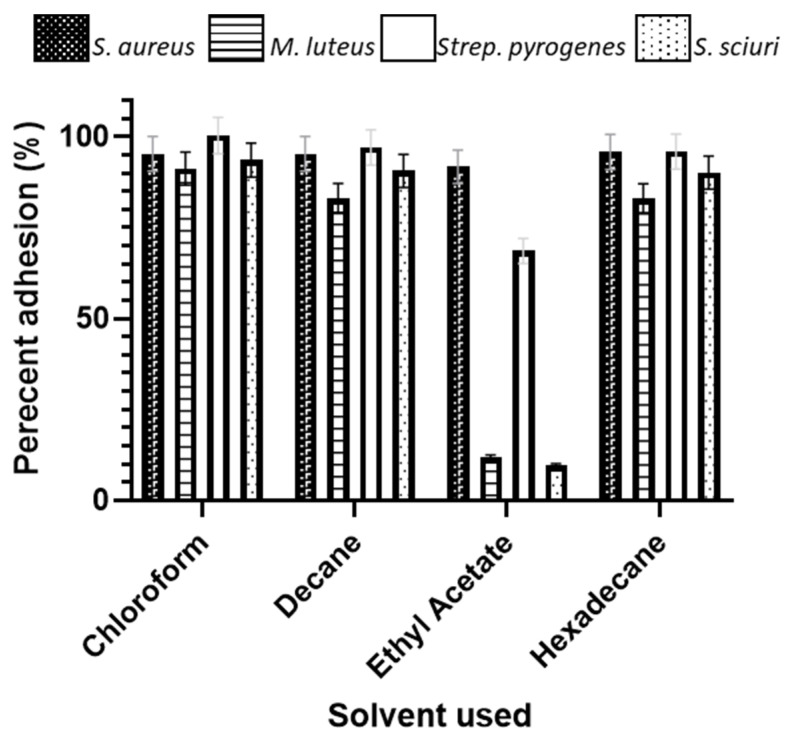
Microbial adhesion to hydrocarbon assay demonstrating the affinity for the bacterial species to solvents with different properties.

## Data Availability

Data will be freely available from the authors upon reasonable request.

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
