# Peer review of "Multifractal Analysis to Determine the Effect of Surface Topography on the Distribution, Density, Dispersion and Clustering of Differently Organised Coccal-Shaped Bacteria"

_antibiotics, 2022, doi:10.3390/antibiotics11050551_

Round 1

Reviewer 1 Report

Review „Multifractal analysis to determine the effect of surface topography on the distribution, density, dispersion and clustering of differently organised coccal shaped bacteria”

The present paper is associated with the surface topography and its influence on bacterial retention on a surface. The authors used Staphylococcus sciuri, Streptococcus pyogenes, Micrococcus luteus, and Staphylococcus aureus to determine the influence of surface topography on the bacteria distribution, density, dispersion, and clustering on titanium surfaces. The scientific topic is very interesting, although the authors need to clarify and add a few things before publication.

At first, I would like to point out that the following sentence in the abstract section is hard-to-read: Four coccal-shaped bacteria, Staphylococcus sciuri, Streptococcus pyogenes, ….. dispersion and clustering when retained on titanium surfaces with defined topographies. I would suggest splitting the sentence into two sentences. 

Also, a few of the words in the abstract and introduction section are in italic. Please revise accordingly.

The sections and sub-sections should be numbered according to the template of the journal. Additionally, the authors should unify the image size ( for example, figures 3 and 4) and try to use colored graphs.

The Surface Topography definition is one of the most crucial things that the authors need to clarify and modify in their study. Surface topography refers to both the profile shape and the surface roughness (including the waviness and the asperity or the finish) and is a measurable quantity according to several norms like the following publication. Due to the nature of the surface, I would suggest that the authors should use a confocal or an atomic force microscope and associate the surface characteristics with the quantity that is associated with the surface topography (for example, Sa, Sz) or surface roughness (for example Ra, Rz) by following the relevant norms.

-Surface texture and integrity of electrical discharged machined titanium alloy. Int J Adv Manuf Technol 115, 733–747 (2021). https://doi.org/10.1007/s00170-020-06159-z

In general the paper is well written and very interesting, but the authors need to revise the paper according to the above mentioned comments.

Author Response

The authors would like to thank the reviewers for their insightful comments. All the comments have been addressed below.

The present paper is associated with the surface topography and its influence on bacterial retention on a surface. The authors used Staphylococcus sciuri, Streptococcus pyogenes, Micrococcus luteus, and Staphylococcus aureus to determine the influence of surface topography on the bacteria distribution, density, dispersion, and clustering on titanium surfaces. The scientific topic is very interesting, although the authors need to clarify and add a few things before publication.

At first, I would like to point out that the following sentence in the abstract section is hard-to-read: Four coccal-shaped bacteria, Staphylococcus sciuri, Streptococcus pyogenes, ….. dispersion and clustering when retained on titanium surfaces with defined topographies. I would suggest splitting the sentence into two sentences. 

We have changed this to the following; Four coccal shaped bacteria, Staphylococcus sciuri, Streptococcus pyogenes, Micrococcus luteus and Staphylococcus aureus that organise in different cellular groupings (grape like clusters, tetrad-arranging clusters, short chains, diploid arrangement) were used. These differently grouped cells were used to determine how surface topography affected their distribution, density, dispersion and clustering when retained on titanium surfaces with defined topographies.

Also, a few of the words in the abstract and introduction section are in italic. Please revise accordingly.

The bacterial genus and species are always noted in italic as they are Latin names so we have not changed this.

The sections and sub-sections should be numbered according to the template of the journal. Additionally, the authors should unify the image size ( for example, figures 3 and 4) and try to use colored graphs.

This has been done and all the graphs have been coloured.

The Surface Topography definition is one of the most crucial things that the authors need to clarify and modify in their study. Surface topography refers to both the profile shape and the surface roughness (including the waviness and the asperity or the finish) and is a measurable quantity according to several norms like the following publication. Due to the nature of the surface, I would suggest that the authors should use a confocal or an atomic force microscope and associate the surface characteristics with the quantity that is associated with the surface topography (for example, Sa, Sz) or surface roughness (for example Ra, Rz) by following the relevant norms.

-Surface texture and integrity of electrical discharged machined titanium alloy. Int J Adv Manuf Technol 115, 733–747 (2021). https://doi.org/10.1007/s00170-020-159-z

In the methods section the following has been added:

2.2 Surface Characterisation

The surfaces were characterised in terms of their topography and wettability according to Verran et al. [21]. Atomic force microscopy (Quesant Instruments, CA, USA) was operated in contact mode to image and characterise the topography of the surfaces. Silicon nitride tips with a spring constant of 0.12 N/m were used. The RMS values (defined as the square root of the mean square) were derived from replicate readings (n = 3). Contact angle measurements were using the sessile drop technique with 5 µL of water (Kruss goniometer, France) using HPLC grade water (BDH, Poole, UK) (n = 5).

In the results section we have added;

3.1 Surface characterisation

The shape of the surface features was quantified using AFM to determine the RMS of the samples (Figure 3). It was determine that there was a significant difference in the RMS value between each of the three surfaces used (p >0.001). Following contact angle measurements, there was no significant variation (p < 0.001) in surface wettability was observed, with an average contact angle of 91Ëš being determined for all three samples. (data not shown).

Figure 3. RMS values of the titanium coated surfaces. **** = p < 0.001.

In general the paper is well written and very interesting, but the authors need to revise the paper according to the above mentioned comments.

Thank you for the lovely comments, we have addressed all the revisions suggested.

Reviewer 2 Report

Dear authors, 

The study entitled " Multifractal analysis to determine the effect of surface topography on the distribution, density, dispersion and clustering of differently organised coccal shaped bacteria" presents an investigation about one specific modified-surface and 4 different bacterias, analyzing several outcomes of these bacterias regarding distribution and biofilm formation over the surfaces. The final outcome to analyze the different responses of different bacterias at the same topography it is interesting. However, the authors need to take into consideration that the topography is only one of the properties that can influence bacterial adhesion, distribution, etc. Thus, discussion and clarification about the other modifications relevant for surfaces are missing in this manuscript.

- Major issues: 

1- The characterization of the modified-surfaces are limited. Information of roughness parameters and wettability are essentials in this manuscript, even same if the authors already have published in a previous article.

2- The difference between the two modified-surfaces (1.02 um and 0.59 um) is quite small, how did the authors measure this size?  Again, roughness parameters here are important to understand the bacterial response in the cultures.

3- I didn't find any description of surface sterilization.

4- The manuscript has a huge lack of information about the other surface properties besides topography (wettability, roughness, chemical composition, base material, crystalline phase, etc). Superficial topography is one of the factors that can influence the bacterial response and surface properties should be analyzed in a set for significance. I suggest the authors explore in the discussion section, articles that showed the differences caused by changes in the surface properties. Several articles reveal this information: Examples: 

- DOI: 10.1016/j.archoralbio.2020.104824
-DOI: 10.1016/j.biomaterials.2010.01.071
- DOI: 10.1016/j.cis.2012.06.015

5- The authors need to describe the limitations of this study in the discussion section and conclusion section. The manuscript shows only one specific type of surface and doesn't explore several surface properties. This is quite different from the clinical reality and needs to be described. The results will be totally different with small changes in some of the properties. 

6- Abstract: I don't agree with the last sentence. The authors demonstrate the influence of only one type of surface on bacterial density, distribution, etc. And not surfaces in general. This should be corrected in the entire manuscript. 

Author Response

Reviewer 2

The study entitled " Multifractal analysis to determine the effect of surface topography on the distribution, density, dispersion and clustering of differently organised coccal shaped bacteria" presents an investigation about one specific modified-surface and 4 different bacterias, analyzing several outcomes of these bacterias regarding distribution and biofilm formation over the surfaces. The final outcome to analyze the different responses of different bacterias at the same topography it is interesting. However, the authors need to take into consideration that the topography is only one of the properties that can influence bacterial adhesion, distribution, etc. Thus, discussion and clarification about the other modifications relevant for surfaces are missing in this manuscript.

- Major issues: 

1- The characterization of the modified-surfaces are limited. Information of roughness parameters and wettability are essentials in this manuscript, even same if the authors already have published in a previous article.

We agree with the comment and have included both the RMS data for the surfaces and the surface wettabilities.

2- The difference between the two modified-surfaces (1.02 um and 0.59 um) is quite small, how did the authors measure this size?  Again, roughness parameters here are important to understand the bacterial response in the cultures.

We had used AFM in previous work, but in this paper we have included the RMS values for completeness.

3- I didn't find any description of surface sterilization. The surfaces were produced in a vacuum chamber and were sterile once produced.

We have added the following; The surfaces were removed from the rig and placed into Sterile Petri dishes in which they were stored at room temperature until use

4- The manuscript has a huge lack of information about the other surface properties besides topography (wettability, roughness, chemical composition, base material, crystalline phase, etc). Superficial topography is one of the factors that can influence the bacterial response and surface properties should be analyzed in a set for significance. I suggest the authors explore in the discussion section, articles that showed the differences caused by changes in the surface properties. Several articles reveal this information: Examples: 

- DOI: 10.1016/j.archoralbio.2020.104824
-DOI: 10.1016/j.biomaterials.2010.01.071
- DOI: 10.1016/j.cis.2012.06.015

This has been done – we have added ‘In most assays, different topographies are made on a range of surfaces that result in a variety of surface property combinations, which also results in the surfaces having a range of different chemistries and hydrophobicities [35]. Surfaces with a range of properties have been shown to elicit a range of effects both increasing and decreasing bacterial adhesion. On titanium implant surfaces, it was demonstrated that an Ra value of 0.2 μm and below did not influence the quantity of bacterial adhesion [36]. For example, it has been suggested that rougher surfaces support bacterial colonization [37,38], although results are conflicting. Early biofilm formation by Streptococcus mutans has also been shown to be reduced on conventionally polished surfaces when compared to surfaces polished with a simplified protocol which removed less material removed from the surface, and hence it was concluded that polishing, did not necessarily reduce oral bacterial colonisation [39]. Differences in the bacterial species can also affect their binding to a surface. It was shown that titanium surfaces showed higher Streptococcus sanguinis adhesion on rough surfaces when they were compared to surfaces with a medium and smooth finish, and although there was a trend,  the results were not statistically significant [40]. However, the use of titanium surfaces was found not to influence the quantity of adhering S. epidermidis [40].  Such results demonstrate how the bacterial species can also affect their retention to surfaces. However, it is difficult using such surfaces to determine how specific surface parameters affect bacterial retention directly. Using very controlled surfaces, produced using the same materials and deposition times, as was used in this work, results in such interactions can be investigated.’

5- The authors need to describe the limitations of this study in the discussion section and conclusion section. The manuscript shows only one specific type of surface and doesn't explore several surface properties. This is quite different from the clinical reality and needs to be described. The results will be totally different with small changes in some of the properties. 

We have added ‘The limitations of this study include that we only investigated linear surface features that are of microbial sizes in attempt to understand the topographical effect on microorganisms with different groupings. However, further work to investigate the different chemistries and hydrophobicities will also be of value in future work.’

6- Abstract: I don't agree with the last sentence. The authors demonstrate the influence of only one type of surface on bacterial density, distribution, etc. And not surfaces in general. This should be corrected in the entire manuscript. 

This has been corrected and removed throughout.

Reviewer 3 Report

The work entitled “Multifractal analysis to determine the effect of surface topography on the distribution, density, dispersion and clustering of differently organised coccal shaped bacteria” evaluates the retention of four different coccal-shaped bacterial species, that organise in different cellular groupings, by surfaces with defined chemistry, topography and physicochemistry. The work is well organized and well put together. The information is presented in a way that is very easily understood and the discussion is fluid. The information provided is somewhat new however there are small details that require the authors attention prior to publication:

  • Reduxe the abstract, it is way too long.
  • The novelty of this research is not clear in the introduction.
  • English writing requires improvement.

Overall, the work has merit and should be considered for publication after minor revision.

Author Response

The work entitled “Multifractal analysis to determine the effect of surface topography on the distribution, density, dispersion and clustering of differently organised coccal shaped bacteria” evaluates the retention of four different coccal-shaped bacterial species, that organise in different cellular groupings, by surfaces with defined chemistry, topography and physicochemistry. The work is well organized and well put together. The information is presented in a way that is very easily understood and the discussion is fluid. The information provided is somewhat new however there are small details that require the authors attention prior to publication:

  • Reduxe the abstract, it is way too long.

This has been done.

  • The novelty of this research is not clear in the introduction.

We have added the following ‘. The novelty of this work is that most work evaluates the effect of surface topography based on cellular size, rather than on bacterial groupings.

  • English writing requires improvement.

This has been revised throughout the manuscript.

Overall, the work has merit and should be considered for publication after minor revision.

Round 2

Reviewer 1 Report

The authors have improved the paper's quality, although they still need to improve a couple of the previous comments.

I suggest using the proper surface roughness unit according to the norms. The authors should define the Ra (Roughness Average of a surface measured microscopic peaks and valleys.).

Also, since they used an AF microscope, I would suggest adding the images. 

In general, the paper is well written, and if the authors incorporate the comments mentioned above, the article is ready to get published. 

Author Response

We would like to thank the reviewer for their comments. We have addressed all the comments made.

I suggest using the proper surface roughness unit according to the norms. The authors should define the Ra (Roughness Average of a surface measured microscopic peaks and valleys.).

We have added:  Peaks and troughs are measured using a profilometer, Ra (roughness average) is the arithmetic average of the absolute values of the profile height deviations from the mean. 

Also, since they used an AF microscope, I would suggest adding the images. #

We have added these.

Reviewer 2 Report

The authors corrected and adjusted relevant points in the manuscript. 

Now, in my opinion, the manuscript has the quality for publication. 

Author Response

Thank you for your kind consideration of the manuscript.